# Whole-Genome Identification and Characterization of the DKK Gene Family and Its Transcription Profiles: An Analysis of the Chinese Soft-Shell Turtle (*Pelodiscus sinensis*)

**DOI:** 10.3390/ani14060931

**Published:** 2024-03-18

**Authors:** Yongchang Wang, Junxian Zhu, Chen Chen, Liqin Ji, Xiaoyou Hong, Xiaoli Liu, Haigang Chen, Chengqing Wei, Junjie Zhang, Xinping Zhu, Wei Li

**Affiliations:** 1Key Laboratory of Tropical and Subtropical Fishery Resource Application and Cultivation, Ministry of Agriculture and Rural Affairs, Pearl River Fisheries Research Institute, Chinese Academy of Fishery Sciences, Guangzhou 510380, China; ayongchanga@163.com (Y.W.); zhujunxian_1994@163.com (J.Z.); chenchen@prfri.ac.cn (C.C.); jiliqin@prfri.ac.cn (L.J.); hxy@prfri.ac.cn (X.H.); liuxl@prfri.ac.cn (X.L.); zjchenhaigang@prfri.ac.cn (H.C.); zjweichengqing@prfri.ac.cn (C.W.); zhuxinping@prfri.ac.cn (X.Z.); 2College of Life Sciences, Xinjiang Agricultural University, Urumqi 830052, China

**Keywords:** DKK gene family, *Pelodiscus sinensis*, expression analyses, sex differentiation

## Abstract

**Simple Summary:**

The DKK gene family plays an important role in immune mechanisms and embryonic development, but studies on *Pelodiscus sinensis* have not been reported. In this study, for the first time in the context of *P. sinensis*, we analyzed the relevant features of the DKK gene family using bioinformatics methods and constructed transcriptional profiles and tissue expression profiles. The *DKK1* and *DKK3* genes were found to be highly expressed in the ovaries, while *DKKL1* and *DKK4* were significantly differentiated in the testes. In this study, we preliminarily investigated the function of the DKK gene family in the context of *P. sinensis*, and this study lays the foundation for further systematic research on the molecular mechanism of sex differentiation in *P. sinensis*.

**Abstract:**

The DKK family is a canonical small family of WNT antagonists. Though recent studies have suggested that the DKK gene family may be involved in sex differentiation in *Pelodiscus sinensis*, there are still a lot of things about the DKK gene family that we do not know. In this study, we used bioinformatics methods to identify members of the DKK gene family in *P. sinensis* and analyzed their phylogeny, covariance, gene structure, structural domains, promoter conserved sites, signal peptides, gonadal transcription factors, transcriptional profiles, and tissue expression profiles. Additionally, qRT-PCR results were utilized for the validation and preliminary investigation of the function of the DKK gene family in *P. sinensis*. The results showed that the DKK gene family is divided into six subfamilies, distributed on six different chromosomal scaffolds containing different gene structures and conserved motifs with the same structural domains, and all of the members were secreted proteins. Our transcriptional profiling and embryonic expression analysis showed that *DKKL1* and *DKK4* were significantly expressed in the testes, whereas *DKK1* and *DKK3* were significantly upregulated in the ovaries. This suggests a potential function in sex differentiation in *P. sinensis*. Our results may provide a basic theoretical basis for the sex differentiation process in *P. sinensis*.

## 1. Introduction

The Dickkopf (DKK) genes are a small evolutionarily conserved gene family consisting of four members (*DKK1*-*4*) and a unique *DKK3*-related gene, *DKKL1*, all of which encode secreted proteins with two distinct cysteine-rich structural domains [1]. DKK proteins were first identified in amphibians as endogenous inhibitors of WNT signaling and inducers of head development [2]. Among them, Dickkopf-1 (*DKK1*), a founding member of the family, is a secreted glycoprotein that has been identified as a head inducer and WNT antagonist in early *Xenopus laevis* embryos, and its overexpression induces abnormal cranial development [3]. Its expression was observed to be elevated in *Mus musculus* tissues mediating the epithelial–mesenchymal transition, suggesting that this protein may be involved in cardiac, dental, hair follicle, and limb development, as well as osteoinduction [4]. In *Homo sapiens*, *DKK1*, Dickkopf-2 (*DKK2*), and Dickkopf-4 (*DKK4*) are located in the same homologous chromosome group [5]. All of these genes are able to regulate WNT signaling, and in most cases, their actions are inhibitory. *DKK1*, *DKK2*, and *DKK4* can directly bind to LRP6 and act as WNT antagonists [6]. In addition, *DKK1* and *DKK2* can induce the endocytosis of LRP6, thereby inhibiting WNT–β-catenin signaling [7]. It has recently been found that *DKK2* can also activate the WNT signaling pathway [8,9]. In contrast, *DKK4* has mainly been associated with a large number of *H. sapiens* cancers, including colorectal cancer [10], melanomas [11], hepatocellular carcinomas [12], and gastric cancer [13].

Dickkopf-3 (*DKK3*) was identified after screening a *H. sapiens* brain cDNA library for homology to *Xenopus laevis DKK1*. Compared to the *DKK1* gene, the *DKK3* gene is more strongly expressed in later stages of embryonic development, with *DKK3a* (*DKK3L*) being expressed predominantly in the posterior ganglion, while *DKK3b* (*DKK3*) is expressed in the endocrine pancreas [14]. Like *DKK1*, *DKK3* has mainly been studied in oncology. It has been proposed as a potential tumor suppressor and therapeutic target for several *H. sapiens* cancers [15], and due to the *DKK3* gene being the subject of increasing studies, a gene possessing a sequence homology with *DKK3* has been identified, namely *DKKL1* (Dickkopf-like 1), a distant member of the DKK family that is a secreted glycoprotein with a potential role in spermatogenesis [16]. *DKKL1* has been shown to inhibit the WNT-induced stabilization of β-catenin in many cell types and vertebrate species [17,18]. In adult *M. musculus*, *DKKL1* gene expression was restricted to the testes, and *DKKL1* mRNA was abundantly expressed in developing spermatocytes, first in the developing acrosome and then accumulating in the acrosome of mature spermatozoa [19,20]. By targeting the knockout of the *DKKL1* gene, *M. musculus* embryos develop normally and form fertile offspring [21]. However, the deletion of the *DKKL1* gene results in the severe impairment of in vitro sperm fertilization. In addition, the *DKKL1* gene has been closely associated with the development of weak spermatogenesis [22] and infertility [23] in *H. sapiens*.

The Chinese soft-shell turtle (*Pelodiscus sinensis*) belongs to Reptilia, Testudoformes, Trionychidae, and *Pelodiscus*, and it is an important species of Chinese freshwater aquaculture [24]. The growth rate of male *P. sinensis* exhibits a pronounced sexual dimorphism, increasing at a rate 1.5 times faster compared to that of the females [25]. Therefore, the study of sex differentiation in *P. sinensis* and the discovery of more primary genes involved in sex differentiation are essential for the breeding of all male *P. sinensis*. Previous studies have shown that the sex differentiation of *P. sinensis* is the result of multi-gene regulation. For example, Sun et al. discovered that the gene *Dmrt1* plays a pivotal role in the sexual differentiation of male *P. sinensis*. Furthermore, they observed that the suppression of *Dmrt1* led to a reversal in sex (from male to female) [26]. In a study by Jin et al., the knockdown of *Foxl2* in ZW embryos resulted in the sexual reversal of females to males, along with the significant up-regulation of the testicular markers *Dmrt1* and *Sox9*, whereas the overexpression of *Foxl2* in ZZ embryos resulted in males experiencing a large degree of feminization [27]. Zhang et al. found that *Rspo1* was required for female sex differentiation in Pelodiscus sinensis. *Rspo1* loss of function by RNA interference led to partial female-to-male sex reversal, with masculinized changes in the phenotype of the gonads, the distribution of germ cells, and the expression of testicular regulators [28]. Zhou et al. knocked out the *Amh* gene using RNA interference technology and proved that the *Amh* gene had a necessary and sufficient role in promoting testicular development and spermatogenesis in *P. sinensis* [29]. Recently, a new study involving RNA-Seq found that a member of the DKK gene family, *DKKL1*, was sexually dimorphic in the male and female gonads of *P. sinensis* during early development, and its expression in the testes was significantly higher than that in the ovaries, which implies that *DKKL1* is likely involved in the sex differentiation of *P. sinensis* [30]. However, there is still an extreme lack of relevant studies on the DKK family in *P. sinensis*. In this study, Based on the whole set of transcriptome data available in our group, we screened to obtain the gene sequences of the DKK-able family members [31]. We used bioinformatics to identify the members of the DKK gene family in the *P. sinensis* genome and further analyzed their phylogenetic relationships, covariance analysis, genetic structure, conserved motifs, structural domains, conserved sites in the promoter region, gonadal expression profiles, transcriptional profile, and tissue expression profiles. The findings of this research will enhance our knowledge of DKK genes and establish a theoretical foundation for future comprehensive investigations into the regulatory mechanisms underlying sexual differentiation in *P. sinensis*.

## 2. Materials and Methods

### 2.1. Whole-Genome Identification and a Physicochemical Analysis of the DKK Genes

All genome sequences, protein sequences, and annotation files of *P. sinensis* were obtained from the National Center for Biotechnology Information (NCBI) database in the United States. We downloaded Hidden Markov Model (HMM) files for the DKK gene (IPR039863) from the Pfam Protein Family Database (https://www.ebi.ac.uk/interpro/entry/InterPro/IPR039863, accessed on 10 July 2023). The HMMER 3.0 software [32] was employed to identify members of the DKK gene family within the *P. sinensis* genome. Subsequently, the candidate sequences were validated and redundant ones were eliminated through detailed analysis using the NCBI Conserved Domain tool (https://www.ncbi.nlm.nih.gov/Structure/cdd/wrpsb.cgi, accessed on 17 July 2023) [33] and SMART database (https://smart.embl.de/smart/set_mode.cgi?NORMAL=1, accessed on 20 July 2023) [34]. Finally, the physicochemical properties of the DKK proteins were analyzed using ProtParam (https://web.expasy.org/protparam/, accessed on 20 July 2023) [35].

### 2.2. Analysis of Phylogeny and Synteny within the DKK Gene Family

The amino acid sequences of vertebrate species typically found in P. sinensis, Salmo salar, Danio rerio, Alligator mississippiensis, Xenopus tropicalis, Chrysemys picta bellii, Rana temporaria, Chelonia mydas, H. sapiens, Mus musculus, Mauremys reevesii, and Lacerta agilis were obtained from the NCBI database. Sequence alignment was performed using MUSCLE 5.1 [36], after which unrooted evolutionary trees were constructed using the Jones–Taylor–Thornton (JTT) and GammaDistributed (G) models and the maximum likelihood method with 1000 self-expanding replicates using MEGA 7 software [37], and the rootless evolutionary trees were landscaped using the online website ChiPlot (https://www.chiplot.online/, accessed on 3 August 2023) to beautify the linear evolutionary trees. Utilizing the NCBI genome browser (https://www.ncbi.nlm.nih.gov, accessed on 3 August 2023) and Ensembl (https://www.ensembl.org/index.html, accessed on 4 August 2023), a collinear analysis was conducted across species including H. sapiens, P. sinensis, Alligator mississippiensis, Xenopus tropicalis, and Danio rerio.

### 2.3. Analysis of Gene Structure, Structural Domains, and Promoter Conserved Sites

Gene density files for each scaffold were generated based on genome annotation files, and DKK gene structures were displayed using the TBtools II visualization tool [38]. The amino acid sequences of the DKK genes were uploaded to the MEME Suite database, accessible at http://meme.nbcr.net, accessed on 8 August 2023 [39], for the prediction of conserved motifs, and the number of motifs was examined until the default threshold was exceeded. Meanwhile, the NCBI Conserved Domain tool (http://www.ncbi.nlm.nih.gov/Structure/cdd/cdd.shtml, accessed on 11 August 2023) [33] was utilized to screen the conserved structural domains of DKK proteins with default parameters. The conserved motifs and structural domains of the DKK proteins were visualized using TBtools software [38].

### 2.4. Transcription Factor Analysis and Signal Peptides

A promoter region 2 kb upstream of the transcription start site of the DKK gene family was extracted using TBtools software. Potential transcriptional binding factors in the core region of the promoter were predicted by setting a threshold greater than 90% using the online software PROMO (https://alggen.lsi.upc.es/cgi-bin/promo_v3/promo/promoinit.cgi?dirDB=TF_8.3, accessed on 15 August 2023) and JASPAR (https://jaspar.elixir.no/, accessed on 15 August 2023). Afterwards, the predicted results were subjected to database comparison, and the intersections were filtered to label the transcription factor binding sequences in the promoter core region 2 kb upstream of the DKK gene family of *P. sinensis*, respectively. The prediction of signal peptides was performed using the online prediction website SignalP 5.0 [40].

### 2.5. Sexually Dimorphic Expression Profiling of the DKK Gene Family Based on Transcriptomics Data

The gonadal transcriptome data of *P. sinensis* were downloaded from the SRA database at NCBI SRA accession (PRJNA838782). Valid data (clean reads) were obtained by filtering out low-quality data with splice sequences from raw reads using Trimmomatic 0.39 software [41]. Clean reads were compared to the *P. sinensis* genome using HISAT 2.2.1 [42], after which the default parameters of the String Tie (v2.1.4) software [43] were applied to calculate the expression of the DKK genes and normalized by the number of fragments per kilobase (FPKM) value. After identifying differentially expressed genes, we employed the edgeR package 4.3 [44] to conduct a detailed analysis. Genes were considered significantly differentially expressed if they met a *p*-value less than 0.05. For genes exhibiting high significance, we further applied additional criteria of FDR < 0.05 and |log2 FC| > 1. Subsequently, the normalized expression levels, expressed as log2TPM+1, were visualized in a gene expression heat map using TBtools, allowing us to correlate and compare the expression patterns across genes.

### 2.6. Quantitative Real-Time Fluorescence-Based PCR Analysis of the DKK Gene Family

#### 2.6.1. Sample Collection

To explore the functionality of DKK gene family members, qRT-PCR was utilized to measure their relative expression levels across various tissues of adult male and female ovaries. All experimental methods were conducted in compliance with the Pearl River Fisheries Research Institute’s animal husbandry regulations, located in Guangzhou, China. Three adult male and three adult female *P. sinensis*, all sourced from Huizhou Wealth Xing Industrial Co. (Huizhou, China), were chosen as the subjects for this experiment. The corresponding tissue samples were obtained following the sample collection protocol described by Lei et al. [31] and were promptly preserved in liquid nitrogen for RNA isolation.

#### 2.6.2. RNA Extraction, cDNA Synthesis, and Quantitative Real-Time PCR (qRT-PCR)

Total RNA extraction for all samples was conducted adhering to the guidelines provided by RNAiso Plus (Takara, Beijing, China). The RNA quality was then assessed through RNA electrophoresis on a Bio-Rad PowerPacTM system (Bio-Rad, Hercules, CA, USA) and using a NanoDrop 2000 spectrophotometer (ThermoFisher, NanoDropOne, Waltham, MA, USA). Subsequently, cDNA synthesis was carried out following the specifications outlined in a reverse-transcription kit from Takara (Beijing, China). The qRT-PCR assay was conducted in accordance with the directions provided by the iTaq Universal SYBR Supermix (BIO-RAD, Hercules, CA, USA). All primers were developed based on the nucleotide sequences of the DKK family genes of *P. sinensis* available on NCBI. The *Ef1α* gene, which exhibits a consistent expression pattern in *P. sinensis* [30], was selected as the reference gene for calculating the relative expression of the target gene. Three biological samples were selected for qRT-PCR experiments and three biological replicates were performed for each individual.

The primer sequences for both the target and reference genes are presented in Table 1. The expression levels of transcripts were quantitatively analyzed using the 2^−ΔΔCT^ method [45]. Differential expression analysis was performed by conducting an ANOVA [46]. The results were expressed as the mean ± SEM of three replicates, and statistical significance was established at *p* < 0.05.

## 3. Results

### 3.1. Identification and Physicochemical Characterization of DKK Gene Family Members in the Genome of P. sinensis

Six DKK gene family members were initially identified in the *P. sinensis* genome, namely *DKK1*, *DKK2*, *DKK3*, *DKK3L*, *DKK4*, and *DKKL1*. Our analysis of the physicochemical properties of the proteins showed that the length of the DKK family sequence of *P. sinensis* ranged from 201 to 389, and the molecular weight varied between 22,229.6 and 43,169.58 kDa (Table 2).

### 3.2. Phylogenetic Analysis of the DKK Genes

The phylogenetic relationships of the DKK genes in *P. sinensis* were investigated by constructing a linear phylogenetic tree. The results showed that the DKK genes of *P. sinensis* can be categorized into six subfamilies: *DKK1*, *DKK2*, *DKK3*, *DKK4*, *DKK3L*, and *DKKL1*. The topology of the phylogenetic tree showed that the DKK genes of *P. sinensis* first clustered with turtles, followed by reptiles and mammals, and the DDK genes of *P. sinensis* were distinct from those of fish (Figure 1).

### 3.3. Collinear Analysis

We performed a covariate analysis on *H. sapiens*, *Alligator mississippiensis*, *P. sinensis*, *Xenopus tropicalis*, and *Danio rerio*. The results showed that the *DKK1* gene of *H. sapiens*, *P. sinensis*, and *X. tropicalis* was located downstream of *PRKG1* in all cases, while the *D. rerio DKK1* was upstream of *PRKG1B*, which may be due to the fact that *PRKG1* evolved from *PRKG1B* (Figure 2A). The *DKK2* gene of *P. sinensis* and *X. tropicalis* was located in the chromosome segment SGMS2-PAPSS1-DKK2, whereas that of *H. sapiens* and *P. sinensis* was located in the chromosome segment composed of the genes LEF1-HADH-CYP2U1–SGMS2-PAPSS1-DKK2-GIMD1-AIMP1-TBCK-NPNT-GSTCD (Figure 2B). The *DKK3* gene in *H. sapiens*, *P. sinensis*, *X. tropicalis*, and *D. rerio* was tightly linked to *USP47* in all cases, suggesting that this chromosome segment was highly conserved during evolution (Figure 2C). The POLB-DKK4-VDAC3 chromosome fragment was highly conserved among *H. sapiens*, *P. sinensis*, and *Rhincodon typus*, but *POLB* was replaced by *IKBK3* in *A. mississippiensis* (Figure 2D). The *DKKL1* gene in both *A. mississippiensis* and *P. sinensis* was located upstream of *CCDC155* (Figure 2E).

### 3.4. Gene Structures

In the gene structure analysis, we found significant differences in the gene length, exons, and introns of the DKK gene of *P. sinensis*. The highest number of exons among the six genes was seven (*DKK3*), and the lowest was four (*DKK2*, *DKK3L*, *DKK4*, and *DKKL1*) (Figure 3A). By comparison with the *Homo sapiens* DKK gene, we found that the two genes have identical numbers of exons and introns, even though their gene lengths are different. It was predicted that the DKK gene of *P. sinensis* is highly conserved in vertebrates (Figure 3B).

### 3.5. Prediction of Structural Domains and Signal Peptides of the DKK Family

DKK1, DKK2, DKK3L, and DKK4 all contain two cysteine-rich structural domains (CRD1 and 2), and Sgy is uniquely present in DKK3 and DKKL1, demonstrating their unique homology (Figure 4). Our signal peptide prediction of *P. sinensis* DKK proteins showed that all six proteins have signal peptides, indicating that they are all secreted proteins (Figure 5A–F). Among them, DKK3 had the longest signal peptide length (57 amino acids; Figure 5C), and DKK4 had the shortest signal peptide length (18 amino acids; Figure 5E).

### 3.6. Transcription Factor Predictions and Conserved Promoter Loci of the DKK Gene Family

We performed a prediction analysis of transcription factors, and we forecast that 548 transcription factors interacted with the promoter region of the 2 kb region upstream of the 5′ of the DKK genes in *P. sinensis*. We selected the top 10 transcription factors for visualization (Figure 6) and found that *PRDM9* and *ZNF281* were the most common transcription factors, with zinc finger proteins being the most widely distributed. Not only that, transcription factors related to sex differentiation (*RARA*, *FOXL2*, *DMRT1*, *SOX9*, *SRY*, *SOX17*, *PAX2*, and *TBX1*) were also predicted.

In addition, we performed predictive analyses of conserved sites in the promoters of the DKK gene family in turtles and tortoises. We found that the DKK gene promoter region has a large number of identical conserved sites in all three species. Among them, *DKK1* has more identical conserved sites, such as *DKK1*: “TGGAAAGTTT”, “ACAAGGCCCGATCCTGCCTCTAAT”, and “CCTAACAGAGCCG”. In contrast, *DKKL1* and *DKK3L* have fewer identical conserved sites, such as *DKKL1*: “GATTGG” and “GCTTGT”; and DKK3L: “CAGGCCTGG”. It suggests that *DKK1* is strongly conserved in turtles and tortoises and that the gene functions may be similar (Figure 7).

### 3.7. Gonadal Transcriptional Profiling

To explore the transcriptional patterns of DKK genes in the male and female gonads of *P. sinensis*, we conducted an analysis of transcriptomic datasets derived from the early stages of gonad development in this species. *DKK1* and *DKK3* were highly expressed in the ovaries, *DKKL1* was highly expressed in the testes, and *DKK4* was minimally expressed in the testes. There were extremely low expression levels of *DKK2* and *DKK3L* in the ovaries and testes (Figure 8).

### 3.8. Expression Profiling of the DKK Genes in Different Tissues of P. sinensis

To further corroborate the transcriptome findings and investigate the expression patterns of DKK genes across various adult tissues of *P. sinensis*, we conducted additional studies. We performed qRT-PCR on six DKK genes in the heart, brain, kidney, spleen, liver, testis, and ovary tissues of *P. sinensis*. The results showed that there were significant differences in the expression patterns of these six DKK genes. *DKKL1* was highly expressed in the testes and significantly higher than in other tissues, consistent with the transcriptome results. While *DKK1* was highly expressed in the spleen and *DKK4* and *DKK3* were highly expressed in the heart, their expression in the ovaries and testes was significantly different. Both *DKK2* and *DKK3L* were highly expressed in the spleen (Figure 9).

## 4. Discussion

In the present study, we successfully identified all members of the DKK gene family within the *P. sinensis* genome, revealing that they possess a conserved structural domain. Nevertheless, notable variations were observed in terms of gene structures, amino acid sequences, and conserved motifs, indicating potential functional distinctions among these genes. The results of our gene structure analysis showed significant differences in intron–exon structure and motif composition among different DKK subfamilies, suggesting that DKK family members may be functionally diverse. For example, *DKKL1* had only the Sgy structural domain, and its exon length was significantly different from the other subfamilies. Tissue expression profiling showed that *DKKL1* was highly expressed in the testes of *P. sinensis*, which was significantly different from the expression pattern of other DKK gene subfamilies. The conserved regions of the DKK gene family and the results of our covariance analysis showed that the genes neighboring the DKK family genes can be highly conserved in *H. sapiens* and *P. sinensis*, except for *DKKL1* and *DKK3L*. This indicates a possibility that they might be governed by comparable mechanisms and execute analogous functions across both species. Phylogenetic relationship analysis can provide important evidence for the classification and evolution of gene families. In a phylogenetic tree, the closer the relatives in the evolutionary tree, the more similar the gene structures are [47]. All protein sequences were categorized into six subfamilies based on the evolutionary tree, and all subfamily members, except for the *DKKL1* gene, were highly conserved among the different species.

Transcription factors have the ability to bind to cis-acting elements, thereby modulating gene expression and playing a crucial role in developmental processes [48]. The WNT signaling pathway has been shown to be associated with testicular and ovarian development [49,50,51]. For example, high expression levels of *WNT11* and *WNT9b* were found in granulosa cells and oocytes of *Oncorhynchus mykiss*, suggesting that they are involved in folliculogenesis and oogenesis in bryozoans [52]. Typical WNT signaling was transduced to the β-catenin protein signaling cascade via the frizzled (FZD) family of receptors and LRP5/LRP6 co-receptors [53,54]. In contrast, DKK family members can interact with and trigger endocytosis of the LRP5/LRP6 co-receptor to prevent the formation of the WNT-FZD-LRP5/LRP6 complex used for typical WNT signaling [55]. For example, in *H. sapiens*, the gonadal cell line *TP53* was able to inhibit the activity of the typical WNT signaling pathway through the up-regulation of *DKK1* expression, thereby affecting gonadal differentiation [56]. The epigenetic silencing of *DKK3* disrupts normal WNT/β-catenin protein signaling and apoptosis regulation [57]. Therefore, the DKK family of genes in *P. sinensis* is likely to affect gonadal differentiation by negatively regulating the WNT signaling pathway. Among the 548 transcription factors in the promoter region of the 2 kb region 5′ upstream of the DKK genes in *P. sinensis*, a large number of transcription factors related to sex differentiation were predicted. *FOXL2* and *DMRT1* are key members of the feminization and masculinization pathways, respectively [58]. A decrease in *SOX9* expression in *H. sapiens* and *M. musculus* leads to XY female development, while its activation in XX embryonic gonads leads to male development [59]. *SOX17* is a key factor in SV flap formation and ST healthy spermatogenesis in *M. musculus* (the seminiferous tubules (ST) in the mammalian testes are connected to the testes (RT) by a supporting valve (SV). Sperm produced in the ST are released into the luminal fluid and passively translocated to the RT via the SV) [60]. In mammals, the master switch regulating the testis-determining pathway is the *SRY* (sex-determining region Y) gene on the Y chromosome [61,62,63]. *SRY* directly up-regulates *SOX9* (SRY-BOX 9) expression, triggering a complex genetic network cascade that mediates testicular differentiation [64,65]. *PAX2* can affect the ovarian development process in *Silurus asotus* directly or indirectly through the WNT signaling pathway. *TBX1* can interact with the retinoic acid (RA) signaling pathway and has been shown to be involved in the control of *O. mykiss* gonadal differentiation [65,66]. Therefore, it is likely that some members of the DKK gene family control sex differentiation in *P. sinensis* by interacting with the above transcription factors.

The expression patterns of genes are known to be intricately linked to their functional roles [67]. The analysis of gene expression patterns in different tissues can provide a basic reference for predicting the functions of some unknown functional genes. For the first time in *P. sinensis*, we analyzed the expression pattern of the DKK family, and we found that *DKKL1* was highly expressed in the testes, *DKK3* was highly expressed in the heart, and *DKK1*, *DKK2*, *DKK3L*, and *DKK4* were all highly expressed in the spleen. These findings are similar to those of some previous studies: Yan et al. found that the expression of the *DKKL1* gene was restricted to the adult *M. musculus* testis and that *DKKL1* mRNA was abundantly expressed in developing spermatocytes, first in the developing acrosome and subsequently accumulating in the acrosome of mature spermatozoa [68]. Not only that, Kohn et al. found that *DKKL1* is also an N-glycosylated protein involved in sperm synthesis during spermatocyte maturation [21]. Piek et al. reported that *DKK3* exhibits a notably elevated expression level in the heart tissues and secreted proteins of *H. sapiens*, indicating a relatively high absolute cardiac expression [69]. In reptiles, although the spleen has some hematopoietic function, the immune function it plays in the organism is more important [70]. Although functional studies of *DKK1*, *DKK2*, *DKK3L*, and *DKK4* in the spleen are rare, all their main functions are understood to be related to immune aspects [71,72,73]. Interestingly, although *DKK1* and *DKK3* were highly expressed in the spleen and heart, respectively, we found that their expression in the ovaries was significantly higher than in the testes. Some studies suggest that the WNT signaling pathway plays an important role in ovarian development and hormone secretion [74,75]. *DKK1* has an important role in the WNT signaling pathway and is involved in embryonic development [76,77]. Although *DKK1* has been reported to play a role in the normal development of *M. musculus* testes [78], *DKK1* has been found to be up-regulated in the ovaries of *Shovelnose Sturgeon*, and some researchers have hypothesized its involvement in oocyte maintenance, folliculogenesis, and oogenesis [79]. *DKK3*, as a soluble WNT inhibitor that can positively or negatively regulate the WNT signaling pathway, is involved in embryonic development. It has been shown that *DKK3* function and expression can influence ovarian development [75]. In *Muscovy ducks*, *DKK3* can interact with *HTRA3* and *RSPO3* to affect their ovarian differentiation [80]. Similarly, although *DKK4* was highly expressed in the spleen, we found its expression to be significantly differentiated from the ovaries and testes. Previous studies have shown that WNT/β-catenin signaling is required for the designation of primordial germ cells and the normal development of the male fetal reproductive tract and that the WNT signaling pathway in the testes specifically contributes to the proliferation of SSC and progenitor cells [81,82]. Takase et al. found that the WNT inhibitor *DKK4* is highly expressed in the canalicular compartment in mice. *DKK4* blocks the activation of WNT/β-catenin protein signaling in meiotic spermatocytes, spermatids, and the lumen, where spermatozoa are located [83]. Therefore, *DKK1*, *DKK3*, *DKK4*, and *DKKL1* may be involved in the sex differentiation process and may regulate the related functions in *P. sinensis*.

## 5. Conclusions

In conclusion, we have effectively recognized all components of the DKK gene family within *P. sinensis*. Transcription profiling and tissue expression profiling revealed that *DKKL1* and *DKK4* may be associated with testicular development and spermatogenesis in male *P. sinensis*. In addition to their involvement in autoimmunity and heart formation in *P. sinensis*, *DKK1* and *DKK3* may also be related to oocyte maintenance, folliculogenesis, and oogenesis in females.

## Figures and Tables

**Figure 1 animals-14-00931-f001:**
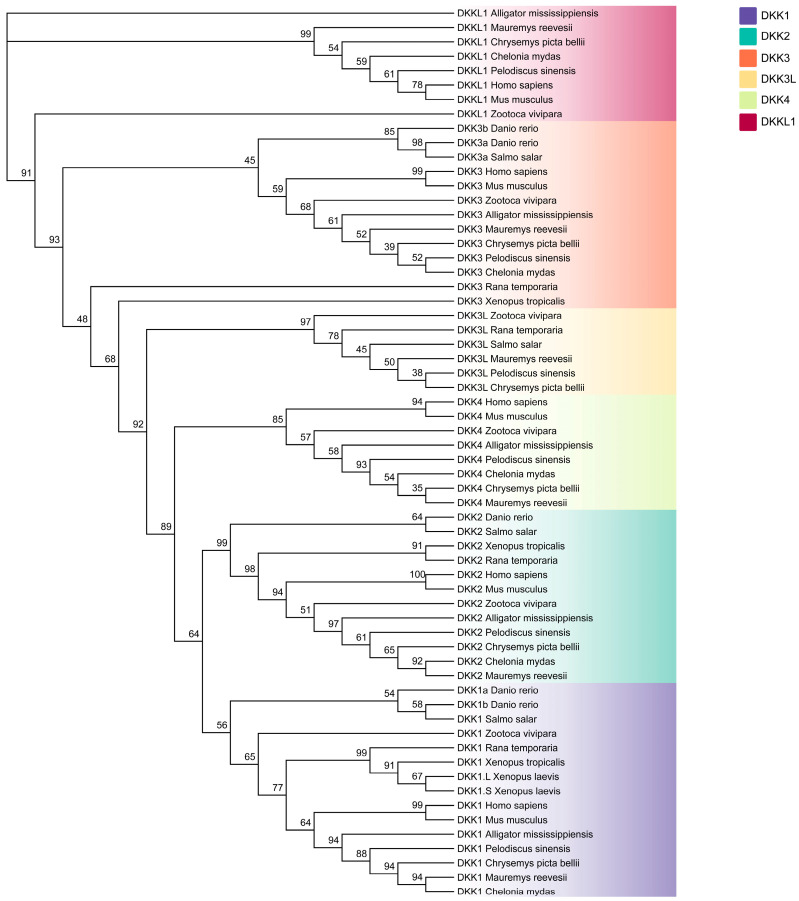
The phylogenetic tree illustrates the relationship among DKK gene family members across different vertebrate species. Distinct colors are used to designate various DKK gene subfamilies.

**Figure 2 animals-14-00931-f002:**
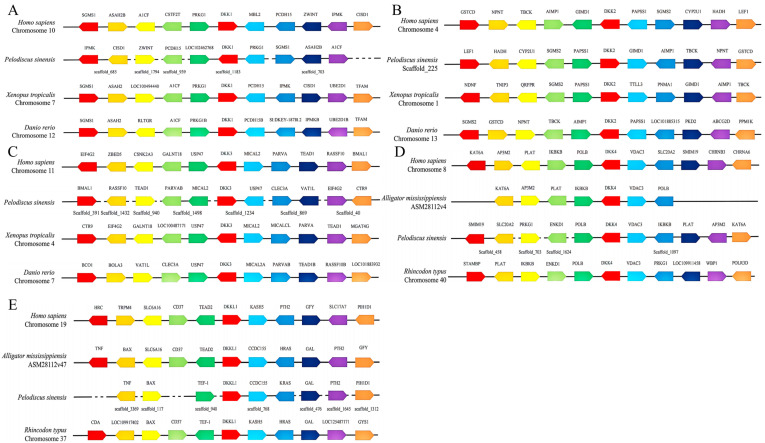
Collinear analysis of *DKK1* (**A**), *DKK2* (**B**), *DKK3* (**C**), *DKK4* (**D**), *DKKL1* (**E**), and their neighboring genes in *H. sapiens*, *Alligator mississippiensis*, *P. sinensis*, *Xenopus tropicalis*, and *Danio rerio*. The direction of the arrow indicates the direction of transcription, and each solid line represents a chromosome scaffold.

**Figure 3 animals-14-00931-f003:**
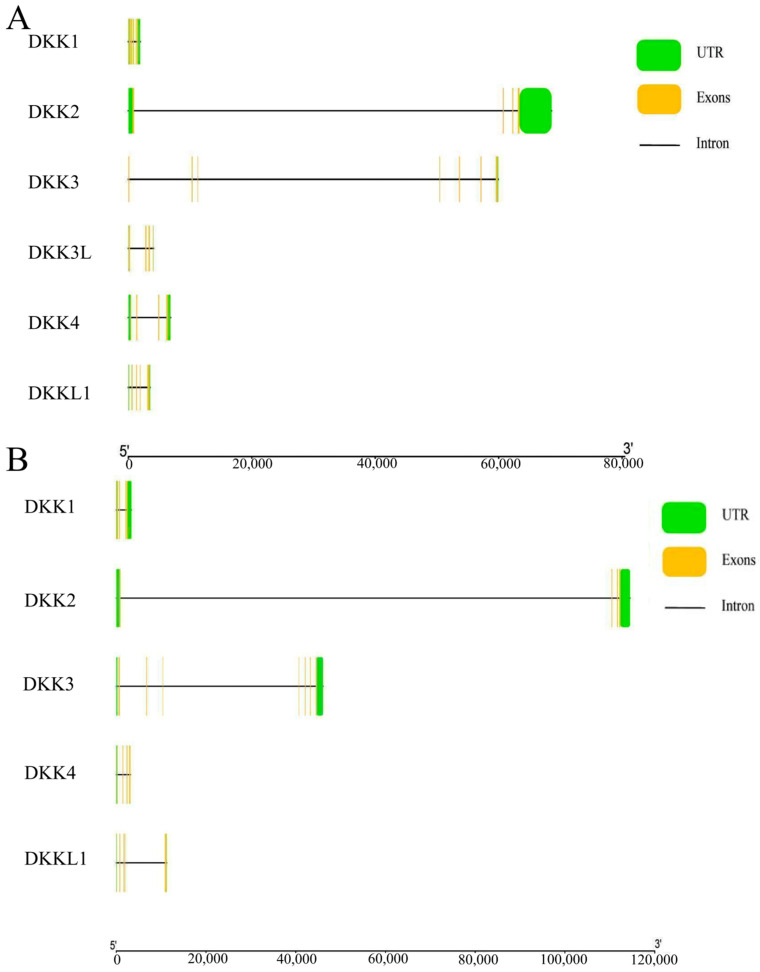
(**A**) Gene structure of the DKK gene in *P. sinensis*. (**B**) Gene structure of *Homo sapiens* DKK gene. Green boxes, yellow boxes, and black lines, respectively, indicate non-translated regions, exons, and introns.

**Figure 4 animals-14-00931-f004:**
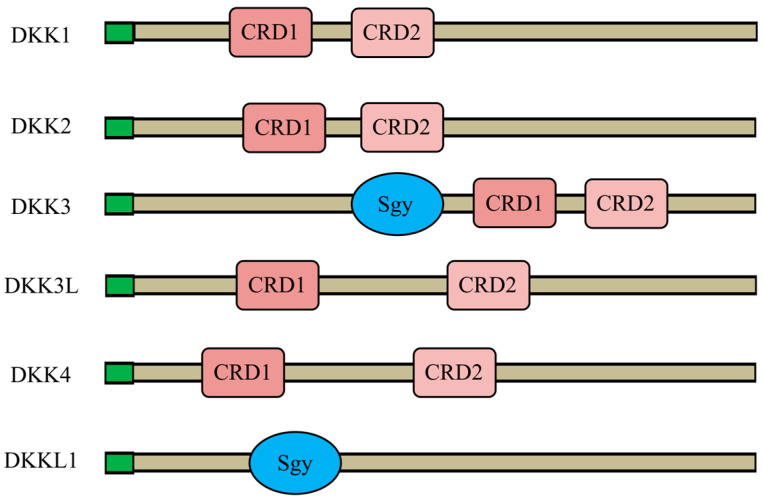
DKK family structural domains. Green boxes, N-terminal signal peptides; Sgy, Soggy domain; CRD, cysteine-rich domain. Uniprot ID: DKK1, K7GBU1; DKK2, K7FG86; DKK3, K7FZM3; DKK4, K7FAY0.

**Figure 5 animals-14-00931-f005:**
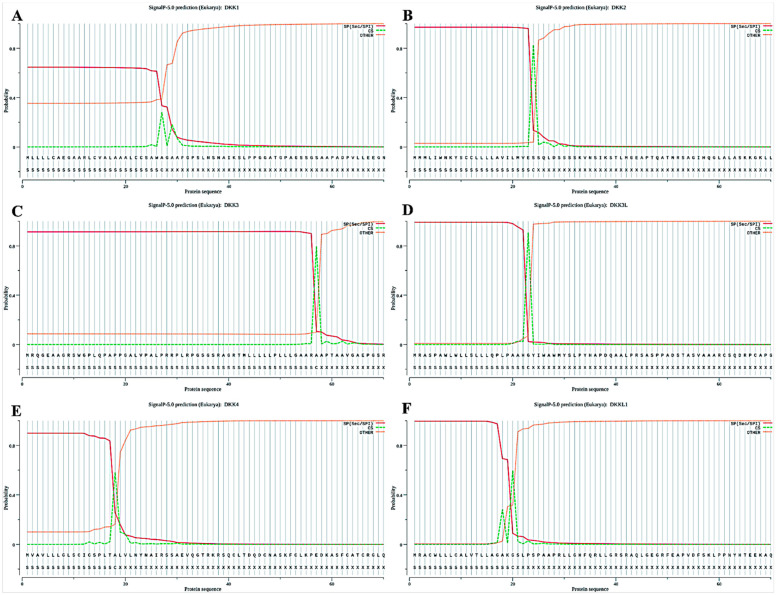
Signal peptide predictions for the DKK proteins. The figures (**A**–**F**) show the signal peptide prediction results for DKK1, DKK2, DKK3, DKK3L, DKK4 and DKKL1, respectively. The forecasting tool is SignalP 5.0, an online forecasting website.

**Figure 6 animals-14-00931-f006:**
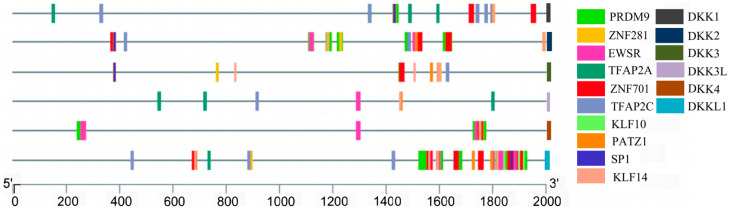
Analysis of the DKK gene promoters in *P. sinensis.* The different colored rectangles indicate the different transcription factors. The prediction tools used were the TBtools software and the online software PROMO (https://alggen.lsi.upc.es/cgi-bin/promo_v3/promo/promoinit.cgi?dirDB=TF_8.3, accessed on 15 August 2023) and JASPAR (https://jaspar.elixir.no/, accessed on 15 August 2023).

**Figure 7 animals-14-00931-f007:**
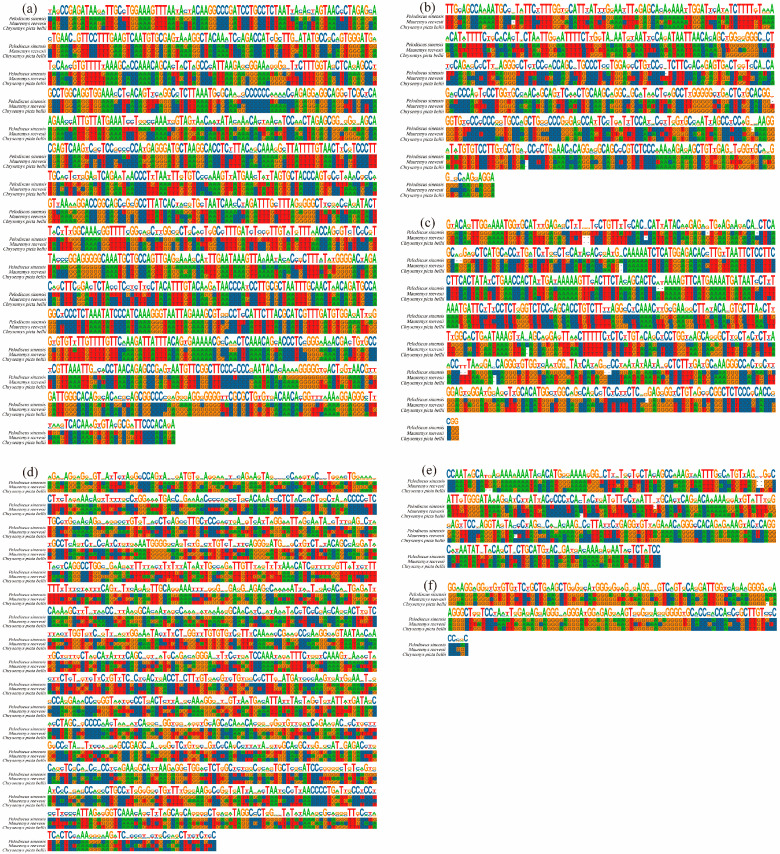
Conserved promoter loci of the DKK gene family in *P. sinensis*, *Mauremys reevesii,* and *Chrysemyspictabellii*. (**a**) DKK1; (**b**) DKK2; (**c**) DKK3; (**d**) DKK3L; (**e**) DKK4; and (**f**) DKK1L.

**Figure 8 animals-14-00931-f008:**
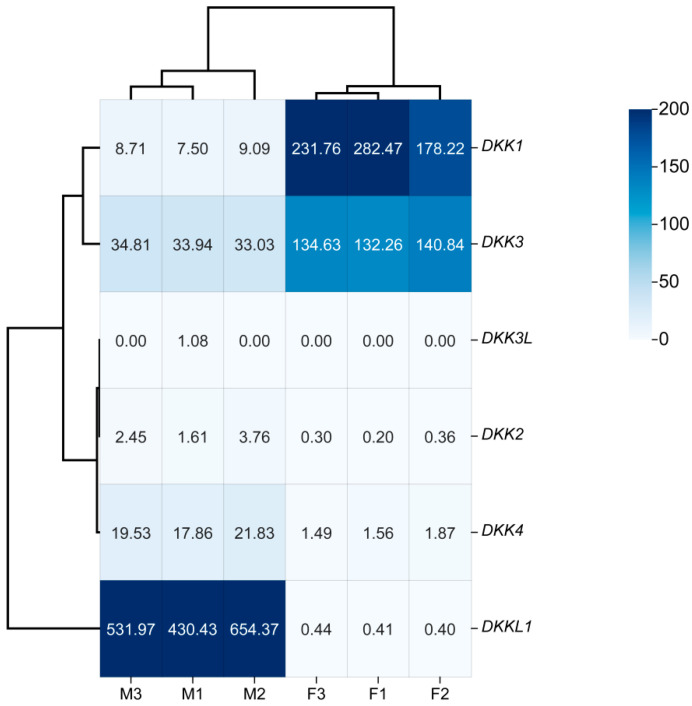
Expression analysis of DKK genes in *P. sinensis.* The expression levels of the DKK genes are indicated by FPKM values. The color range is 0~200. Blue indicates high expression, while white indicates low expression. F1, ovary 1; F2, ovary 2; F3, ovary 3; M1, testis 1; M2, testis 2; M3, testis 3.

**Figure 9 animals-14-00931-f009:**
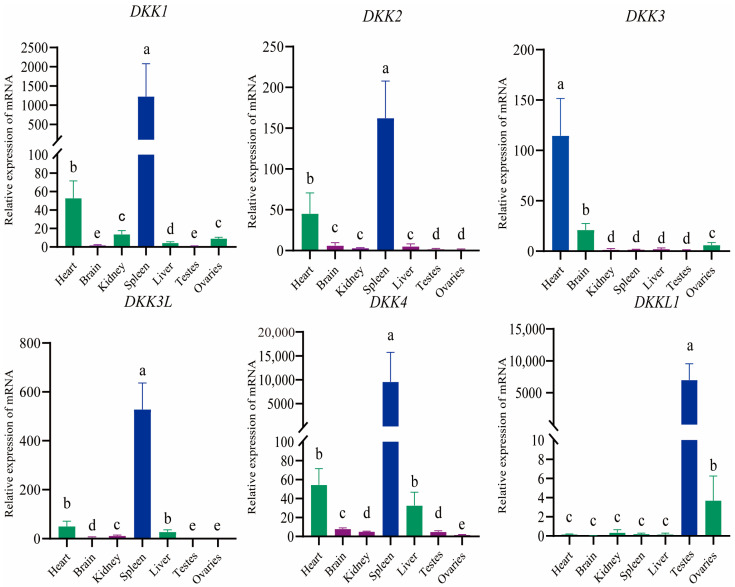
Analysis of the expression levels of DKK genes in different tissues of *P. sinensis* using qRT-PCR. The *efα1* gene was used as an internal reference. Means shared with different superscripts differ significantly (*p* < 0.05), differences labeled with the same lowercase letter are not significant, and differences labeled with different lowercase letters are significant.

**Table 1 animals-14-00931-t001:** The primer sequences of our qRT-PCR analysis.

Primer Name	Sequence (5′–3′)	Product Size (bp)
DKK1-F DKK1-R	CCTCAACTCCAACGCTATCAA	123
	ACGGGCTGGTGCTTGTTA	
DKK2-F DKK2-R	ATCGGCAAGGAGAGGCATAC	130
	TCTCTGTGGCAACGCTTCTT	
DKK3-F DKK3-R	GGAGGAGGCGAGTCTGA	181
	TTTGGTGTCCGTGTTGG	
DKK3L-F DKK3L-R	TGTATTCGCTGCCCTACCAC	240
	TGTGGCACTGGCCAAACATA	
DKK4-F DKK4-R	GCGTTCCTGAAGAATGGT ACTCCTGATGGCGTTGTAG	97
DKKL1-F DKKL1-R	ATGGCTAGCAGCCTGTGTCT GACCTGGCAAAGAGATGGAG	230
Ef1α-F Ef1α-R	ACTCGTCCAACTGACAAGCCTC CACGGCGAACATCTTTCACAG	337

**Table 2 animals-14-00931-t002:** DKK protein analysis of *P. sinensis*.

Name	Gene ID	Number of Amino Acid	Molecular Weight
DKK1	XM_006130932.3	261	28,025.89
DKK2	XM_006119304.3	268	29,923.9
DKK3	XM_006113024.2	389	43,169.58
DKK3L	XM_025186202.1	201	22,229.6
DKK4	XM_006113233.3	218	23,646.1
DKKL1	XM_014572465.2	222	24,848.67

## Data Availability

All datasets generated or analyzed during this study are included in the published article.

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
