# Peer review of "Whole-Genome Identification and Characterization of the DKK Gene Family and Its Transcription Profiles: An Analysis of the Chinese Soft-Shell Turtle (Pelodiscus sinensis)"

_animals, 2024, doi:10.3390/ani14060931_

Round 1

Reviewer 1 Report

Comments and Suggestions for Authors

Comments and Suggestions for Authors

The authors identified and characterized the DKK gene family and transcription profiles of the Chinese soft-shell turtle (Pelodiscus sinensis) based on data obtained from NCBI. Their findings suggested a potential function of DKK genes in sex differentiation in P. sinensis. However, the authors need to be more careful with over interpreting these results based solely on data analysis. Future work is necessary to establish the physiological function of DKK genes in the early developing gonads. Below you can find my comments:

1.     According to the authors' statement, DDKL1 exhibits differential expression in the early developing gonads of P. sinensis, suggesting its potential association with sex differentiation (Line97-101). In section 3.8, the authors also analyze transcriptome data from early developmental stages. However, it is confused that the authors ultimately chose adult male and female individuals to validate transcriptional expression. It is advisable to select early developing gonads of both males and females, as this represents the critical period of sex differentiation. In addition, the number of individuals collected and the number of biological replicates performed during the qRT-PCR experiment were not mentioned in the manuscript.

2.     Figure2. To me, the resolution of map is low and the words are blurry and hard to see. Similar issues exist in other figures such as Figure 3 and Figure 6, so I am unable to provide an evaluation for this part.

3.     Line32-34. “Our transcriptional profiling and tissue expression profiling results showed that the expression of DKKL1 and DKK4 in the testis differed significantly from that of the ovaries, whereas DKK1 and DKK3 were significantly up-regulated in the ovaries” The expression of this sentence is not clear, please rewrite it.

4.     Line97-101. No corresponding references are available here.

5.     Line92 and line 122. Pelodiscus sinensis should be in italics. Check through manuscript for such mistakes.

6.     Line169-170. Log2 FC should be log2 FC.

7.     Line 282. Punctuation error.

8.     Figure 9. Spelling mistake: “efα1”. Check through manuscript for such mistakes.

9.     Line 335. Grammatical mistake: “they and”

10.  Line389-390. Grammatical mistake “DKK3 is specifically highly expressed in H. sapiens heart tissues and secreted proteins”. Define “HF”.

11.  Add discussion about DKK protein structure and subcellular localization.

12.  Please double check references. There is an error in the format of reference 10. Liang, J.; Sun, L.; Li, Y.; Liu, W.; Li, D.; Chen, P.; Wang, X.; Hui, J.; Zhou, J.; Liu, H.; et al. Wnt signaling modulator DKK4 inhibits colorectal cancer metastasis through an AKT/Wnt/β-catenin negative feedback pathway. The Journal of biological chemistry 2022, 298, 102545, doi:10.1016/j.jbc.2022.102545, doi:10.1016/j.jbc.2022.102545.

Comments on the Quality of English Language

The manuscript contained errors in writing and grammar, and it was necessary to edit and polish the language.

Reviewer 2 Report

Comments and Suggestions for Authors

Report on Wang et al., “Whole-genome identification and characterization of the DKK 2

gene family …”

Wang et al. describe their analysis of the DKK gene family with a focus on Pelodiscus sinensis, the Chinese soft-shell turtle. They analyze the gene structures, their relationships, predicted protein structure and predicted gene regulation, their gene expression pattern, as well as other features.

Such an analysis is welcome, even though one may wonder why the focus is on this particular turtle and how the reference material was selected.

Overall, the analysis is well supported although not always well documented. For instance, the authors often make statements matter-of-factly although they have only predicted them, without any experimental confirmation. This needs to be stated very explicitly (see below for details). Besides that, I would like to suggest a number of corrections and changes, as follows.

Introduction

First, the authors should mention where the sequence / genome is coming from. I don’t see a single citation of the original genome sequence paper (Wang et al. 2013, doi:10.1038/ng.2615).

Methods

I also suggest to add a table summarizing the database accession numbers of the genomes, genes and proteins used for Figures 1 and 2. This study is not reproducible without these details.

Even though Wang et al. cite 2 papers for the “unique nutritional value [24] and medicinal value [25]” of Pelodiscus sinensis, I very much doubt that there is anything unique in its nutritional value or its medicinal utility. I would remove such questionable references, even though I do not doubt that the economic value may be in the billions (trade with rhino horn is in the billions too despite its absolute medicinal uselessness).

Line 97: authors say that “a new study involving RNA-Seq found that a member of the DKK gene family, DKKL1, was sexually dimorphic …” but no such study is cited. Please add a reference.

Section 2.5. Transcription factor analysis

Please fix the URLs: http:∥ www. Genomatix. de/cgibin∥ matinspector) and

JASPAR (http:∥jaspar. genereg. net/, i.e. remove spaces and add proper slashes (//).

Section 2.7.2. RNA extraction, …

Explain why the Ef1a gene was chosen as the reference gene.

Results

Line 202: add amino acids after 389

Lines 203-205: I suggest to remove the theoretical isoelectric point (Theoretical pI in Table 2), the “instability coefficient” (or index) which is not explained anywhere, including the methods section; same for the lipid solubility coefficient (called Aliphatic Index in Table 2) and the average hydrophilicity coefficient (called Grand Average of Hydropathicity in Table 2).

These parameters do not appear to be really relevant in this context. These coefficients may be more relevant in a structural or biochemical review of the whole family, as opposed to the family members in a single turtle species.

It may be a good idea to add some size ranges for the proteins in Table 2 (as they have been used already in Figure 1), just to indicate whether the proteins are in the right range, which is relevant as many genome sequences actually are truncated due to incomplete sequencing or assemblies.

Table 2: amino acids instead of amino acid.

Figure 1: personally, I tend to use linear trees (as opposed to circular). The names are much easier to read. That said, I would make the colors much lighter; when printed, this tree may be difficult to read as many printers make the colors too dark.

3.3. Collinear analysis

Figure 2: the gene names in the reviewer’s pdf have a resolution that’s too low to read. Equally important, Wang et al. should comment on the missing genes, especially in the scaffolds 1183, 1234, 768 (or 758 ?), and 1097. I wonder if the genes outside the DKK clusters are simply missing from the genome assembly, or if they are simply on different scaffolds (which should be added, but with a gap, so that it becomes clear that they MAY be colinear, if a more complete sequence becomes available). At least it should be indicated whether the sequence is incomplete, e.g. using dashed lines or a size indicator (how long as those Pelodiscus scaffolds ?).

In the legend, I would say “the arrow indicates the direction of transcription” (instead of gene direction).

3.4. Chromosome scaffolding localization and gene structures

While I agree that Figure 2 is very useful and interesting, Figure 3A is not. I would simply delete it, given that very similar information is already provided in Figure 2 (in fact, if Fig. 2 was extended as suggested, there would be no information lost). In any case, the neighboring genes on scaffolds 1183 and 1234 can and should be shown in Fig. 2 as this would be interesting on its own.

Figure 3B is quite useful. However, I would modify it in two ways: first, I would stretch DKK2, so it takes the whole width of the figure. This would make the exons clearer (I would call them exons, not CDS – the latter is often used for the whole CDS). In fact, I suggest to add the gene structure from, say, the human DKK homologs (as a representative of a high-quality reference genome). I would expect that most, if not all, exons are conserved, so it would be nice to see this across vertebrates.

Figure 3, legend, line 250. There is no such thing as a “50k genome. I would say 50k segment or window or something.

Line 244: as above, use exon instead of CDS

Line 246: “DKK3 lacked the 5' non-coding region (5' UTR) (Figure 3B).” – I suspect this is a sequencing or assembly error, or possibly alternative splicing or low-conservation exon. Please verify using the genome sequence of some related reptile or other genome.

Figure 4. This figure is unsatisfactory. I would delete part A – I don’t even know what is shown here. In part B, I would show the domains that are conserved in other model organism DKK proteins, and use the same nomenclature. For instance, some papers call the domains in DKK1,2,4 CRD1 and CRD2 (for cysteine-rich domain, e.g. Kikuchi et al. 2021, https://doi.org/10.1016/j.semcdb.2021.11.003). In fact, this figure may be a good opportunity to include the Uniprot IDs, so that the proteins can be found easily (Uniprot isn’t mentioned anywhere in the paper). BTW – I just realize that Uniprot has actually 2 DKK1 sequences from P. sinensis (K7GBU1 and K7GBS6), hence it’s even more important to identify which one is meant (I haven’t checked the other DKK proteins in Uniprot, but the authors should).

3.6. DKK protein structure, subcellular localization, and signal peptides

Please emphasize that these are not experimental but predicted localizations and structures.

Lines 275 ff – please replace extracell with something like extracellular or secreted. I have never heard the word “extracell”.

Table 3. I assume these percentages refer to the whole protein sequence, please say so in the legend. In any case, I think these statistics are not very useful., unless done for the whole family. I would delete it.

Figure 5 may be combined with Figure 4, so it would put those domains into some context. Maybe also indicate functions in Figure 4, if known, e.g. if there are known and biologically relevant and thus confirmed binding partners, otherwise the structures are also floating around in the manuscript without much meaning. In any case, it should be explicitly stated in the legend that these are predicted structures (how ? Alphafold?).

Figure 6. I tend to include the predicted signal peptides in Figure 4 and relegate Figure 6 to a supplement. Anyway, the current version of the figure is too low resolution so that barely anything can be recognized. If used, include the method (prediction tool) in the legend.

3.7. Transcription factor predictions

Line 287: Wang et al. claim that they “found that 548 transcription factors interacted with the promoter region”. No, they did NOT. They predicted TF binding sites, which is completely legitimate, but they have to say that explicitly. (statement is repeated in lines 362-364).

Figure 7: please add how predictions were made. I would actually suggest to indicate those TFs that have been experimentally shown to bind to other DKK promoters, or sites that are conserved across turtles or reptiles. This would dramatically improve the value of this figure, even though it is possibly a lot of work. TF binding sites are some of the fastest evolving functional sites, so it’s important to make such predictions, but it’s much much harder to prove that they are functional. BTW – I assume the DKK ORF is on the right-hand end of the lines, but please indicate if that’s correct (possibly putting the protein names in the ORF symbols).

Figure 8: I would add the labels right into the figure, i.e. where it says “M3
 at the bottom I would say “testis 3” (and please then explain in the legend what the difference is between testis 1,2 and 3.

Figure 9. Does efa1 as reference mean it was set to 1?

Line 323: domain > domains ?

Line 325: poor stability is predicted, not demonstrated.

Line 368: explain SV flap and ST spermatogenesis

Line 374: spell out RA (I assume it stands for retinoic acid but Tbx1 is rarely mention in the context of RA).

Line 391: HF ?

Minor issues: in several instances, Pelodiscus sinensis is formatted without italics (in others with), so please add italics to all those cases (e.g. p. 2, line 92, p. 3, line 122 etc), plus many other species in the same paragraph (2.2.). Other species names are in italics but should be in regular font, e.g. Shovelnose sturgeon or Muscovy ducks)

Similarly, some authors are in Sentence case, and others are in uppercase (e.g. YAN et al., line 384, PIEK in line 389)

Comments on the Quality of English Language

Fine; just a few minor typos and other small issues.

Round 2

Reviewer 1 Report

Comments and Suggestions for Authors

Thank you for the revised version of your manuscript entitled "Whole-genome identification and characterization of the DKK gene family and its transcription profiles: an analysis of the Chinese soft-shell turtle (Pelodiscus sinensis)". I appreciate the effort you have put into addressing my concerns and making the necessary revisions. After carefully evaluating your revised manuscript, I think that the improvements made have significantly enhanced the quality and clarity of your work.